# 3D sedimentary architecture showing the inception of an Ice Age

H. Løseth [1✉], J. A. Dowdeswell [2], C. L. Batchelor[2,3] & D. Ottesen[4]

Northeast Atlantic climate shifted into the Quaternary Ice Age around 2.6 M yr ago. Until now, however, the detailed changes associated with this inception of an Ice Age have remained obscure. New high-quality three-dimensional seismic data reveal a detailed geological record of buried surfaces, landforms and sedimentary architecture over vast parts of the Norwegian North Sea. Here, we show the sequence of near-coast geological events spanning the Northeast Atlantic inception of an Ice Age. We identify the location of immediate pre-glacial fluvially derived sandy systems where rivers from the Norwegian mainland built marine deltas. The stratigraphic position of a large submarine channel, formed by enhanced meltwater from initial build-up of local glaciers, is also shown. Finally, we document the transition to full ice-sheet growth over Scandinavia from the ice sheet's earliest position to the later pattern of debris-flow lobes reaching the present-day shelf edge.

---

[1] Equinor ASA, N-7005 Trondheim, Norway. [2] Scott Polar Research Institute, University of Cambridge, Cambridge CB2 1ER, UK. [3] Department of Geoscience and Petroleum, Norwegian University of Science and Technology, N-7491 Trondheim, Norway. [4] Geological Survey of Norway, N-7040 Trondheim, Norway. ✉email: heloe@equinor.com

A dramatic climate shift led the Northeast Atlantic into an Ice Age around 2.6 M yr ago at the beginning of the Quaternary Period. In deep-ocean sediments this transition is recorded as increasing volumes of iceberg-rafted debris (IRD) delivered to the surrounding seas as ice sheets built up over the adjacent mid-latitude land masses[1,2]. Until now, however, the detailed changes in continental-margin sedimentary architecture associated with this transition to an "icehouse" world have remained poorly understood. The recent availability of large blocks of high-resolution industry three-dimensional (3D) seismic data now allows imaging of the sediments and landforms that record this major environmental shift and enables inferences on changing geological processes to be made.

The landscape of the Northeast Atlantic as we know it today was largely established during the mid- and late-Miocene, to be overprinted by subsequent Quaternary glaciation[3]. Huge compressive stresses uplifted Norway and the British Isles and their flanking shelves were also raised and exposed, with a subsiding basin forming between[4–8]. The East Shetland Platform was a major clastic source-area for the northern North Sea Basin. The sands making up the Utsira Formation on the western flank were produced during the Late Miocene to Pliocene[9–16] ("Methods"). At about the same time, sandy sediments from Norway formed the Utsira Formation on the basin's eastern side (Fig. 1a). Here, the Utsira Formation east is bounded at its base by the Mid-Miocene Unconformity (MMU), an unconformity on the entire eastern basin margin that dips smoothly westwards showing reflection truncation, onlap and downlap[17,18]. The basin flanks were flooded, possibly related to termination of the compression phase[8], around the end of the Miocene, but the uplifted Norwegian mountains remained as nucleation points for the build-up of glaciers.

The Utsira Formation is overlain by the down-lapping prograding shelf clinoform sediments of the Quaternary Naust Formation that are often in the form of glacigenic debris-flows[13,19–21]. Naust units A–D record the west to northwestward progradation of these sediments through the Quaternary[13,14,20]. To the west of the basin, an eastward-prograding deltaic unit from the East Shetland Platform continued to build up during the early Quaternary[13,14]. The Norwegian Channel Ice Stream, which drained much of the southern Scandinavian Ice Sheet during successive mid- and late-Quaternary glaciations, eroded parts of both the Utsira and Naust formations and formed an Upper Regional Unconformity (URU) at the base of the Norwegian Channel[13,22–24].

We use a large (35,410 km²) 3D seismic survey obtained from the northern North Sea between Norway and the Shetland Islands (60–62 °N; Fig. 1a) to investigate the geological record spanning the change from relatively warm and fluvially dominated sediment delivery from land to sea during the Neogene, to a Quaternary environment influenced largely by erosion, transport and deposition from glaciers and ice sheets. A new generation of high-quality 3D broad-band seismic data provides continuous coverage of large areas of the Norwegian margin, allowing spectacular imaging of palaeo-surfaces and buried landforms within the late Neogene and Quaternary sediments compared with traditional two-dimensional seismic-reflection data and smaller 3D seismic cubes. Here, we report our observations and interpretation of these data to reveal the sedimentary architecture relating to the pre-glacial and early glacial evolution into the icehouse world of the Quaternary.

## Results
**Pre-glacial fluvial activity.** Our 3D seismic imagery from the northern North Sea show three phases of the changing geological record across the transition from a fluvially dominated Neogene into the glacially dominated sedimentary system of the Quaternary Ice Age (Fig. 1).

The sands of the eastern Utsira Formation are located immediately below the oldest glacial sediments of the Quaternary Naust Formation (Fig. 1c). The sand is up to 200 m thick and covers about 1500 km² on the eastern margin of the northern North Sea Basin (Fig. 1b). Sedimentological evidence from wells (Figs. 1b, 2a) show fine to very coarse sand, pebbles and poor to well sorted sands. Seismic records indicate that the sand unit has a well-defined top and base but internally is acoustically chaotic where RMS (Root Mean Square)-amplitude maps image numerous anastomosing and bifurcating channels (Fig. 2a). Individual channels are usually less than 800 m wide, become narrower to the west, and can be traced for up to 30 km. The channel patterns indicate a west to northwest flow direction. The channel configuration suggests a source near the location of the modern mouth of Sognefjorden, a 200 km long and 1300 m deep fjord system cut into the mountainous interior of Norway (Fig. 1a). The easternmost preserved sand is about 50 km west of the present-day mouth of Sognefjorden (Fig. 1b), noting that erosion by the mid- to late-Quaternary Norwegian Channel Ice Stream has removed Utsira Formation sediments further east.

The geomorphic pattern of the channels preserved within the Utsira Formation east is interpreted to represent a fluvially derived sandy deltaic system. Rivers from the pre-glacial Norwegian mainland, appearing to come from a point source at the mouth of what is now Sognefjorden, contributed sorted sediments to build marine deltas. The buried sandy channel system imaged and interpreted from 3D seismic data in Fig. 2a represents channels eroded into delta foresets; the topset beds and part of the foresets have been removed by subsequent erosion.

**Glacifluvial activity—Sunnfjord channel.** The sinuous Sunnfjord channel[25–27] is cut into the northern portion of the eastern Utsira Formation and was therefore formed after the sand was deposited. The channel is 100 km long, up to 2 km wide and 150 m deep within the 3D dataset (Fig. 2b) but continues north and out of the study area at 62 °N (Fig. 1a). The present dip of the middle and outer channel is ~0.5° but increases to 1.3° in the uppermost 25 km. Near its upper end several sub-channels are present that merge downstream. The original eastern portions of the channels are missing through erosion but the preserved extremities point inshore towards the modern Sunnfjord and Nordfjord (Figs. 1b, 2b). The sediment fill of this channel system comprises a 120 m-thick unit. The lower intra-channel part onlaps the base up-dip. This unit is not dated but the relative age is younger than the eastern Utsira Formation and older than the upper Naust Unit A of Ottesen et al.[14]. Further up-dip the channel is infilled by the west to northwest prograding shelf sediments of the Naust Formation, the oldest of which belong to Naust Unit A. Prograding clinoforms from overlying Naust Formation sediments also fill up the remaining portion of the Sunnfjord channel. Whether the Sunnfjord channel is an immediate pre-Quaternary or early Quaternary event chronologically remains unresolved.

We interpret this major Sunnfjord channel, whose preserved upper arms point towards the Sunnfjord and Nordfjord systems on mainland Norway (Fig. 1a), as a submarine product of enhanced summer seasonal meltwater and sediment delivery after the initial build-up of glaciers and ice caps in the Norwegian mountains. The prograding shelf sediments correlating to the intra-Naust Unit A of Ottesen et al.[14] began the process of infilling the Sunnfjord channel (Fig. 1c).

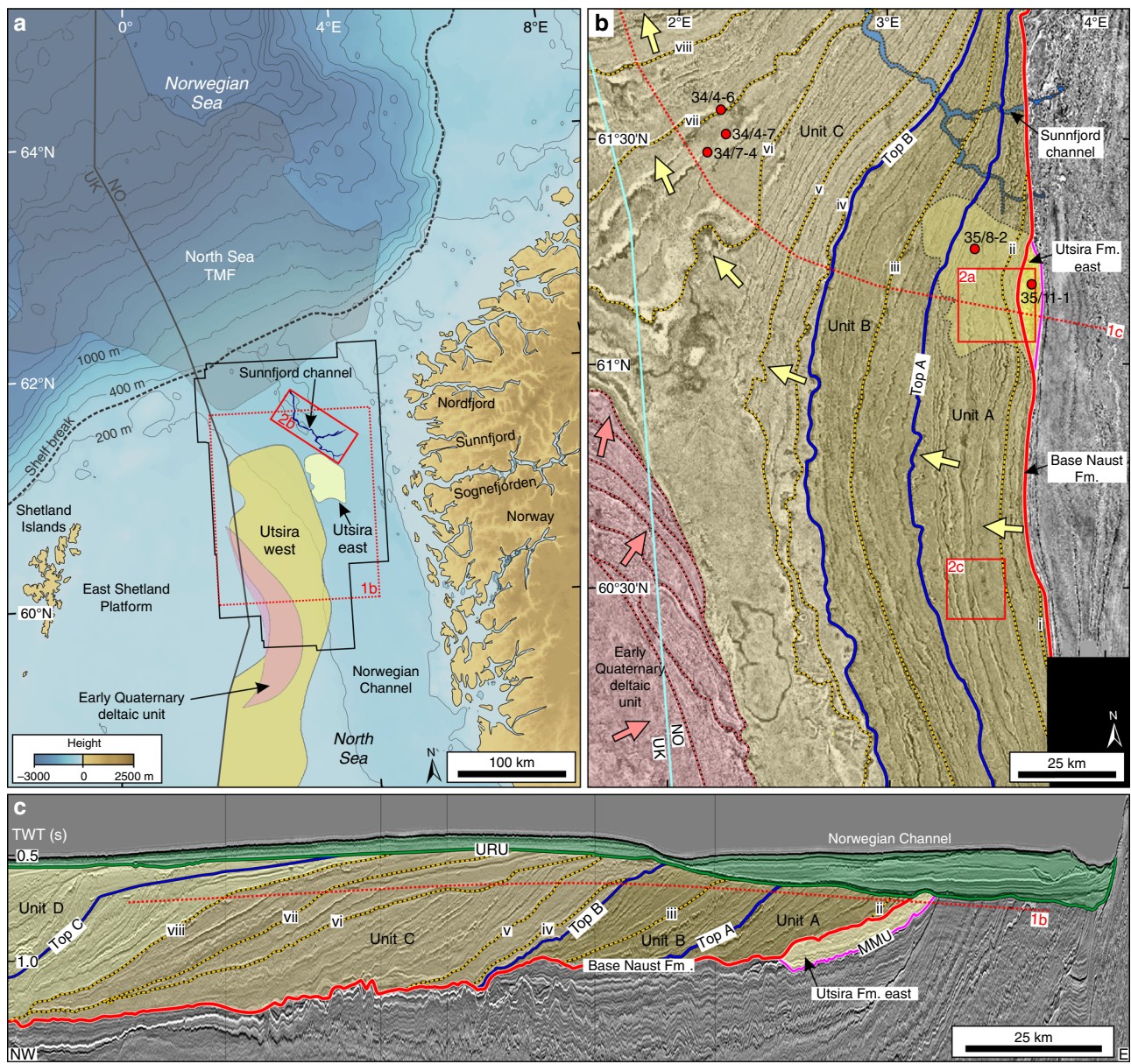

**Fig. 1 Location and seismic data. a** Location map of the northern North Sea margin, showing the position of the 3D time-migrated CGGNVG broadband (BroadSeis) seismic cube (solid black outline). Black dashed line is present-day shelf break. Grey fill is main depocentre of the North Sea trough-mouth fan (TMF) from Nygård et al.[29]. The distribution of the western Utsira Formation sands is from Eidvin et al.[31]. **b** Amplitude map showing the Pliocene to Quaternary infill of the northern North Sea Basin (location in Fig. 1a). The amplitudes are extracted from an arbitrary surface that cuts prograding clinoforms (dashed red line in Fig. 1c). Brown filled area is glacigenic debris-flows that prograded to the west and northwest from the Norwegian mainland. Some of the clinoforms are labelled (i–viii, from oldest to youngest), and the adjusted tops of units A to C of Ottesen et al.[13,14] and Batchelor et al.[20] are shown. Regional ties and estimates suggest the following ages: top Unit A ~2.2–2.0 M yr, top Unit B ~1.7–1.6 M yr, clinoform (v) ~1.5 M yr, and clinoform (vi) ~1.2 M yr (Methods; Supplementary Fig. 1). Pink fill is an early Quaternary deltaic unit that prograded to the northeast from the East Shetland Platform[14]. Yellow fill is the Utsira Formation east (of probable Pliocene age), which was deposited prior to the glacigenic debris-flows. Red circles are wells. **c** Seismic profile showing the Pliocene to Quaternary infill of the northern North Sea Basin (location in Fig. 1b). The top of Unit C is the base of the North Sea TMF. Green fill shows flat-lying units above the Upper Regional Unconformity (URU) in the Norwegian Channel. Colours and labels are the same as in Fig. 1b. Mid Miocene Unconformity (MMU).

**Full ice-sheet glaciation–Naust Formation.** We mapped details of the prograding shelf clinoforms of the Quaternary Naust Formation for the entire northern North Sea from 3D seismic data. This was performed by extracting amplitude values along a detailed constructed surface that intersects the prograding clinoforms (Fig. 1c). In the eastern North Sea Basin, the upper parts of the Naust Formation clinoforms are eroded by the URU and this surface cuts gradually deeper into the eastern Utsira Formation, the MMU, older units and finally into basement.

The positions of one clinoform-intersection appear as a continuous reflection-line in Fig. 1b, showing the location of that clinoform at this time. The detailed positions of all the Naust Formation clinoforms in this map describe the early Quaternary sediment infill of the northern North Sea, like growth rings in trees (we count 47), from the oldest clinoform (i) in the east to the youngest (viii) in the northwest. The east and northeastward migrations of the East Shetland Platform clinoforms are also shown in Fig. 1b. The eastward and westward prograding

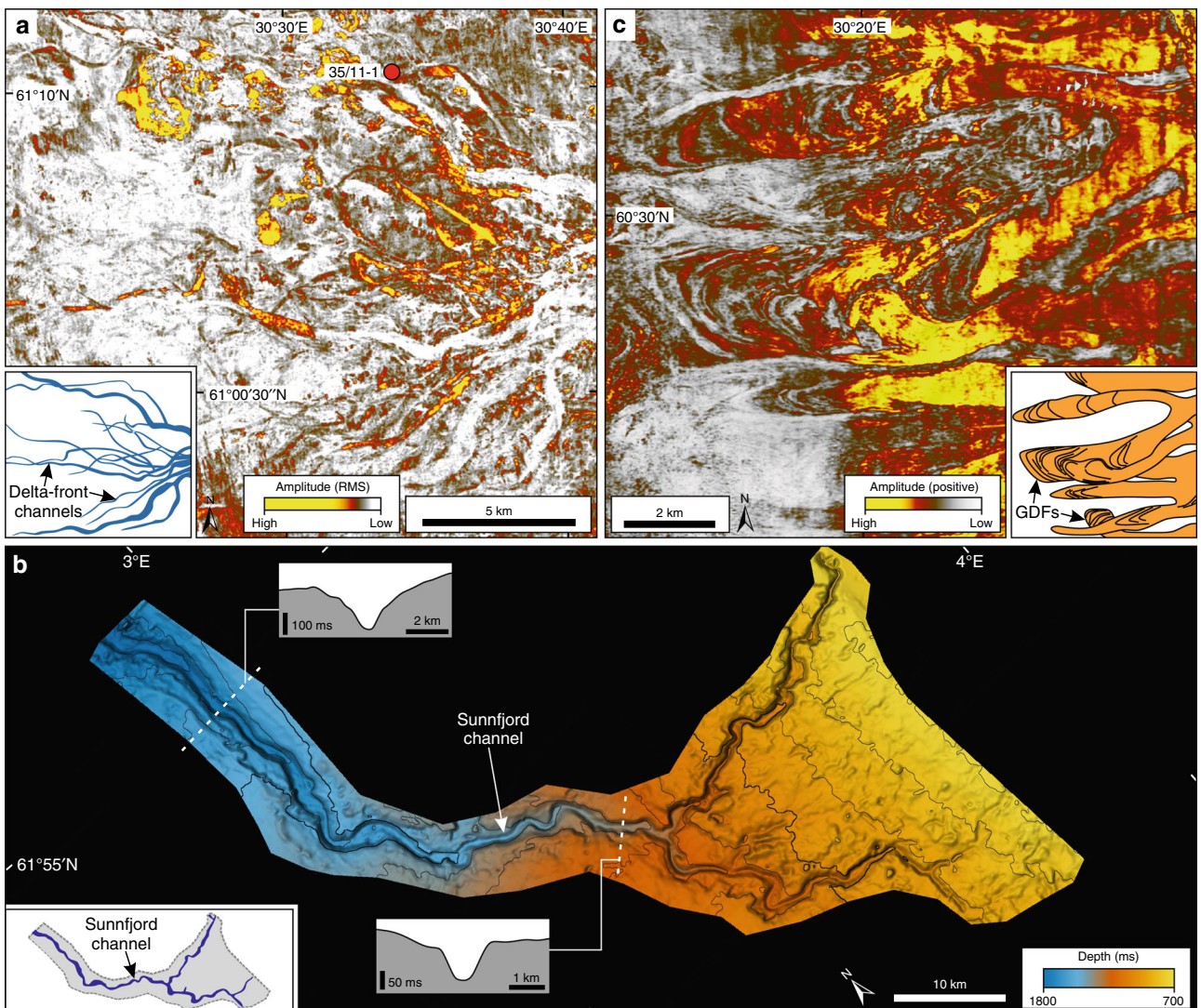

**Fig. 2 Examples (main panels) and interpretations (insets) of geomorphological features extracted from 3D seismic data. a** RMS-amplitude map of delta-front channels in the Utsira Formation east (location in Fig. 1b). Amplitudes are extracted from a 12 ms window around a smooth intra-Utsira east surface where high amplitude values are shown in red and yellow. **b** Structure map of the base of the 100 km-long submarine Sunnfjord channel (c.i. = 100 ms) (location in Fig. 1a). The channel defines the base Naust horizon (base-Quaternary) and is mainly infilled by early Quaternary glacigenic debris-flows. **c** Maximum amplitude map of glacigenic debris-flows (GDFs) on early Quaternary clinoform (ii) (location in Fig. 1b). Maximum amplitudes are extracted from a window 30–10 ms above surface (ii) (red and yellow are high positive amplitude values).

clinoforms meet just after clinoform (v), closing the southward deep marine connection to the central North Sea Basin (Fig. 1b).

Amplitude maps extracted from individual clinoforms in the lower part of the Naust Formation show geomorphic features in the form of depositional lobes (Fig. 2c). These are the plan-form view of the prograding clinoforms found in seismic sections (Fig. 1c). The oldest identified depositional lobes are immediately above the base of the Naust Formation (i) surface, and are probably of earliest Quaternary age ("Methods"). Unfortunately, only the lowest part of the originally assumed 300–400 m high shelf clinoforms is preserved southwest of the mouth of Sognefjorden. Much of the debris making up the prograding clinoforms is fine-grained diamicton, as shown by well-log data from Naust Unit C (Fig. 1b). The prograding Naust Formation surfaces base-lap onto either the top of the MMU, the Utsira Formation, or a lag of older Quaternary sediments (Fig. 1c).

The geomorphic pattern of depositional lobes, imaged on amplitude maps, is interpreted as downslope mass-flow complexes that are typical of many Arctic and Antarctic glacier-influenced prograding shelves. They relate to the rapid delivery of debris from ice streams or outlet glaciers that reach the shelf edge, where the sediment subsequently fails on the upper slope to produce lobate mass-wasting features that are commonly referred to as glacigenic debris-flows[28,29]. The diamictic sedimentology of the glacigenic debris-flows is also similar to that reported from well-logs in Naust Formation Unit C[29]. The presence of these debris flows and prograding clinoforms implies that ice reached the palaeo-shelf edge to deliver sediment from a line-source that began the major build-out of the shelf west of Norway. We locate the oldest preserved Naust Formation sediments in the northern North Sea Basin southwest of the mouth of Sognefjorden, suggesting that an outlet glacier of an ice cap in mountainous Norway was located here during the earliest Quaternary (Fig. 3b). Indeed, this limited glaciation of the shelf is likely to have taken place at a similar time to the development of the glacifluvial Sunnfjord channel further north. The oldest Naust A surfaces have an arcuate shape with radial debris flows indicating an early narrow outlet of ice that spread into an arc on reaching the

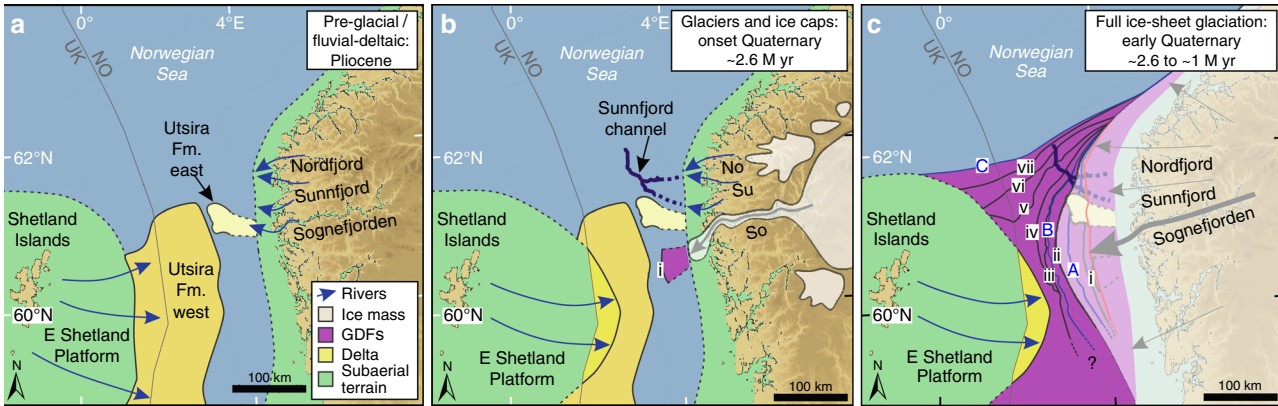

**Fig. 3 Schematic models of the evolution of the northern North Sea margin during the inception of the Quaternary Ice Age. a** During the Pliocene, the sand-rich Utsira Formation west accumulated at the western side of the northern North Sea Basin, fed by rivers that drained across the East Shetland Platform. A smaller delta, the Utsira Formation east, built out beyond Sognefjorden. **b** Around the onset of the Quaternary Period, glacigenic debris-flows record the first advance of grounded ice to the palaeo-shelf break beyond Sognefjorden. Sunnfjord channel was probably formed during this time, and an early Quaternary fluvial delta built out beyond the East Shetland Platform[14]. **c** During the early Quaternary, the northern North Sea Basin was infilled by sediments derived from an ice sheet over southern Norway. The geometry of the clinoform units shows that the Sognefjorden area continued to be the dominant pathway for the transport of glacial sediments to the northern North Sea through the early Quaternary. A delta, fed by rivers that crossed the East Shetland Platform, continued to build up at the western side of the northern North Sea Basin. Topography colours and labels are the same as in Fig. 1. GDF = glacigenic debris-flow.

relatively flat shelf. Relatively limited glaciation of the shelf during the earliest Quaternary was followed by gradually more widespread glaciation after Naust Formation Unit A time (Figs. 1b, 3c), demonstrating the growth of full ice-sheet conditions, with the Scandinavian Ice Sheet reaching the shelf edge along the entire Norwegian margin during mid- and late-Quaternary glaciations[24,30].

The time correlative of the Naust Formation on the western flank of the northern North Sea Basin is an early Quaternary deltaic unit which lies above the western Utsira Formation and is sourced from the East Shetland Platform (Fig. 1). The delta is dated using IRD occurrence in wells[31] and by seismic correlation because the shelf clinoforms of this unit interfinger in the basin with the Naust Formation shelf clinoforms sourced from the Norwegian mainland. The geomorphic pattern of river incision on the outer part of the East Shetland Platform, and channels and fans beyond the shelf edge, suggest that rivers supplied sediments to a marine shelf here. There are no indications of inland ice reaching the East Shetland Platform edge prior to around 0.5 M yr ago[32].

## Discussion
The use of newly available high-resolution 3D seismic data covering a large area of the northern North Sea Basin has allowed us to investigate in detail the changing patterns and processes of sedimentation on this part of the Norwegian margin during the inception of an ice age ~2.6 M yr ago. We present a conceptual model based on this 3D seismic dataset (Fig. 3).

First, in the mid-late Miocene, sedimentation relating to compression and uplift of Norway and the British Isles was initially in the form of fluvially derived sandy deltaic systems represented by the Utsira Formation and older sands (Fig. 3a). Most of these sediments built out into the subsiding northern North Sea Basin from the East Shetland Platform. The upper part of the Utsira Formation, including the Utsira Formation east along the eastern flank, was deposited mainly during the Pliocene after a transgression related to termination of the compression phase[8]. This represents pre-Ice Age fluvio-deltaic sedimentation into a mid-latitude marine basin from the surrounding land masses.

Second, at the beginning of the Quaternary, two lines of evidence indicate initial switching towards an "icehouse"

environment. The cutting of a major channel system, the Sunnfjord channel, into the underlying Utsira Formation sands suggests increasing energy which we link to seasonal glacifluvial activity associated with the initial build-up of glaciers and small ice caps on the Norwegian mainland (Figs. 2b, 3b). At about the same time, the first occurrence of clinoform sediments with a lobate structure, interpreted as glacigenic debris-flows, suggests that ice initially reached a limited area of the palaeo-shelf edge adjacent to the Sognefjorden palaeo-valley, probably in the form of a lobe from a large ice-cap outlet glacier (Figs. 2c, 3b).

Ice-sheet expansion along the Northeast Atlantic margin at the beginning of the Quaternary was driven by large-scale climatic and oceanographic forcing, which was the culmination of a general cooling trend through the Late Cenozoic[33] (Supplementary Fig. 1). Southern Norway was one of the key nucleation centres for the Northern Hemisphere ice sheets during the earliest Quaternary because of its mountainous terrain and proximity to a moisture source[24]. The existence of a deep basin in the northern North Sea during this time[13,14] provided accommodation space for the delivery of large volumes of glacial sediment across the relatively narrow continental shelf to the palaeo-shelf break.

Third, the development of line-sourced clinoforms and lobate debris flows along the eastern edge of the northern North Sea Basin, and later along much of the Norwegian-Barents-Svalbard margin more generally[19,34,35], indicates the onset of full-glacial ice-sheet conditions (Fig. 3c). Major ice-sheet glaciation to the prograding shelf edge then took place a number of times, especially during the glacial-interglacial cycles of the mid-late Quaternary[24,30]. Continued isostatic subsidence of the North Sea Basin through the Quaternary, combined with erosion and isostatic uplift of the surrounding land areas[4-8], facilitated high rates of glacier-derived sediment deposition delivered through expanding glacial valleys. After about 1 M yr ago, the infilling of the North Sea Basin, combined with Northern Hemisphere climatic deterioration and ice-sheet expansion linked to the Mid-Pleistocene Transition[36], enabled confluence of the Scandinavian and British-Irish Ice Sheets, initiation of the Norwegian Channel Ice Stream, and the shifting of the focus of glacigenic sediment deposition into the Northeast Atlantic Ocean[20].

Given that intensive industrial-seismic activity makes the northern North Sea Basin one of the most comprehensively investigated anywhere on Earth, our work provides a high-resolution analogue for the changing sedimentary architecture through similar ice-age inceptions for the earlier ice ages of the last billion years where neither the topographic nor the process setting are likely to be well understood[37].

## Methods

**Seismic dataset**. This study uses a 3D time-migrated CGGNVG broadband (BroadSeis) seismic cube covering 35,410 km² of the northern North Sea, located mainly between 60 °N and 62 °N (Fig. 1a). The inline (N-S) and crossline (E-W) grids are spaced 18.75 m and 12.5 m apart, and data from the seafloor to a depth of 9 s two-way-time were recorded. The vertical sampling interval is 4 milliseconds (ms). The internal velocities of the Quaternary and Neogene sediments are close to 2000 m s$^{-1}$, and a seawater internal velocity of 1480 m s$^{-1}$ was used. The vertical resolution is ~3–5 m in the zone of interest. Seismic horizons were interpreted and extended within the seismic grid using PETREL™ software. The horizons were then gridded to surfaces that formed time-depth maps. Extracted amplitude values, RMS-amplitudes and variance were derived from specified windows around a surface and displayed onto the surface to image geomorphic features (Fig. 2). The original CGG survey did not cover areas around industrial installations, but older 3D cubes were spliced into the data gaps in the final 3D seismic dataset. Seismic interpretation was carried out on the full-stack migrated seismic cube. The final version of the new regional CGG broadband seismic survey was produced in 2018 (Fig. 1a).

**Chronological control**. There is generally poor chronological control on the Neogene and Quaternary sediments of the northern North Sea. Few long sediment cores have been acquired, and biostratigraphic age assignments from industrial wells are problematic because of large numbers of resedimented fossils and a lack of key indicator micro-organisms[38–40]. Here, we provide a summary of available age constraints on the Pliocene to Quaternary sediments of the northern North Sea, based on seismic ties and age estimates from published data. These limited age constraints were used to produce a tentative chronology for the Quaternary sediments of the northern North Sea (Supplementary Fig. 1). This chronology enables the relative ages of the intra-Naust units to be established regionally, and provides a framework into which future age assignments should fit.

The Utsira Formation sand (Fig. 1a) is generally considered to be of Miocene to Pliocene age[15,16,27,31]. It is difficult to assign precise age estimates to the Utsira Formation because of the abundance of resedimented fossils, which makes it difficult to interpret in-situ forms with confidence[39]. In addition, parts of the Utsira Formation have historically been poorly imaged on seismic data because they have been obscured by massive sand injectites[41]. Recent improved age assignments to the Neogene strata of the northern North Sea give slightly younger Pliocene ages for the Utsira Formation[15,16].

The Utsira Formation east is typically assigned an unspecific late Miocene to early Pliocene age in industry completion reports based on biostratigraphic analyses. The stratigraphic position of the Utsira Formation east is above the MMU and below the oldest parts of the Quaternary Naust Formation (Fig. 1). We interpret that the Utsira Formation east sand was deposited after the flooding related to termination of the compression phase[8] and suggest that these sediments are mainly Pliocene in age.

The transition to an icehouse world is documented in cores from the deep-marine sediments in the Vøring Basin on the outer mid-Norwegian margin that show a significant increase in IRD around the Pliocene-Quaternary boundary, implying that glaciers had built up on the Norwegian mainland to reach the open sea by about 2.6 M yr[1,2]. Only one age assignment exists for the early Quaternary sediments in our study area (60–62 °N). The base of the prograding clinoforms of the Naust Formation (i.e., base-Quaternary reflection) has been suggested to be younger than 2.6 M yr and older than 1.4 M yr in this region, based on microfossil investigations of industry wells, including wells 34/7-4 and 34/4-7[15] (Fig. 1b). Although this is a broad time span, it provides a useful minimum age for these sediments.

Further precise intra-Naust Formation ages do not exist in the northern North Sea, and our estimates (Supplementary Fig. 1) are based on seismic ties to the south (Supplementary Fig. 2). The top of Unit B (Fig. 1c) has been suggested to be ~1.7–1.6 M yr in age, based on a seismic tie into the study area from the Netherlands[14]. The top of Unit B is slightly younger than the top of Sequence 13 of Kuhlmann and Wong[42], which corresponds with the top of the Olduvai palaeo-magnetic subchron dated to about 1.8 M yr. This age estimate is in broad agreement with the chronostratigraphic framework developed for the central North Sea by Reinardy et al.[16] (Supplementary Fig. 2), who dated the sediments above the Utsira High using strontium isotopes.

We suggest a tentative age of ~2.2–2.0 M yr for the top of Unit A, because of its stratigraphic position between the base of the Quaternary Naust Formation (~2.6 M yr) and the top of Unit B (1.7–1.6 M yr[14]) (Fig. 1c; Supplementary Figs. 1 and 2). Clinoforms (i) and (ii) within Unit A were therefore probably deposited during the

earliest Quaternary. It is likely that older sediments existed further landward of clinoform (i) but these were eroded by the Norwegian Channel Ice Stream, which formed the URU (Fig. 1c).

Clinoform (v) within Unit C (Fig. 1c; Supplementary Fig. 1) corresponds with horizon R3 of Reinardy et al.[16], which has been dated to ~1.5 M yr (Supplementary Fig. 2). Subsequent to the deposition of clinoform (v), the deep marine basin connection between Norway and the East Shetland Platform closed as a result of sediment infilling from the surrounding land areas (Fig. 1b), preventing reliable seismic correlations to be made between the northern and central North Sea. However, horizon R4 of Reinardy et al.[16], which has been dated to ~1.2 M yr and corresponds to the top of Clinoform Unit 3 of Ottesen et al.[14] (Supplementary Figure 2), can be tentatively correlated to approximately clinoform (vi) by counting the prograding clinoform peak reflections on each side of the shallow basin between the central and northern North Sea. The age of the top of Naust Unit C, which is the base of the North Sea TMF, is uncertain, but it has been suggested to be as young as 0.5 M yr[29].

## Data availability

Shapefiles are available upon request to the first author.

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

## Acknowledgements

We thank Equinor for permission to publish results from internal research and CGG, PGS and TGS for permission to show seismic data. During this work, C.L.B. was funded through the Norwegian VISTA programme.

## Author contributions

H.L. devised the project and interpreted the seismic data with input from D.O. C.L.B. produced the figures with input from all other authors. J.A.D. wrote a first draft of the paper which was improved by input from all other authors.

## Competing interests

The authors declare no competing interests.
