## [Peer Review File · Nature Communications]

Reviewers' comments:

Reviewer #1 (Remarks to the Author):

Review of: Løseth et al. : 3D sedimentary architecture showing the inception of an Ice Age
Nature Communications

This is an interesting paper utilizing a unique 3D seismic data set acquired due to the intense oil and gas exploration in the North Sea. It is nice to see this data set being used for paleoclimatic and sedimentological studies of this kind, and this study brings a more classical geological approach to supplement and improve the more general and standard approaches to derive the ice age history based on deep sea sediment records. The ability to draw on the unique seismic data brings different novel aspects of the development and inception of large-scale Northern Hemisphere glaciation to the table. It is therefore a unique and new contribution and despite elements of the story having been available before in various studies, this is a much more comprehensive and potentially important study. From this point of view I am positive to the paper being published, but have some reservations concerning the contents and its ability to communicate to a wide readership which leads me to suggest a relatively major revision before it can be published.

More specifically:

1. I am not happy with the use of greenhouse/icehouse jargon in this case. Undoubtedly there was a major expansion of NH ice sheets 2.7-2.6Ma, but to call the late Pliocene period immediately before this a greenhouse world is not very accurate, for several reasons: Since there are lots of evidence for and active NH cryosphere long before this time: e.g. the existence of a Greenland Ice Sheet for several million years, and IRD records that point to NH glaciation in the Arctic and Nordic Seas well back into the late Miocene. One may argue that a final step in the emergence of bipolar major glaciations happened around 2.6Ma, but the world had clearly evolved into an icehouse world long before this event. Existing CO₂-records also indicate relatively low atmospheric concentrations leading into the glacial expansion, hence the greenhouse gas forcing had already been driving climate into a world of active cryospheric influence in both hemispheres.

2. The paper deals with the Northern European and Eastern margin of the North Atlantic. Hence, to refer to this as the North Atlantic is misleading, in particular since the timing of glacial expansion may have been different across the North Atlantic, at least with respect to Greenland. The paper should be more precise in referring to the NE North Atlantic or NW European margin.

3. The main weakness of the paper, which also impacts on its readability is the lack of discussion of timing and age control. There is a general lacking discussion of the age of the different units and the underlying age control and its uncertainties. To be published, this needs to be in place including giving numeric values of the time of the events and uncertainties (despite a few dates are indicated in Fig. 3).

It would also help for the general readability if these events were placed into the global and regional deep sea evidence from O-isotopes and IRD-records. A discussion on the possible mechanisms behind the continued expansion of ice sheets/ice streams onto the shelves, e.g. general climatic and oceanographic causes vs changes in ice sheet dynamics due to the changing topography of the mountains and coastal regions due to glacial erosion. Here a mentioning of eustatic and isostatic influences would also be appropriate. I think the paper misses an opportunity to have a wider impact due to these deficiencies.

4. The caption of fig. 3 includes discussion and arguments which should have appeared in the main text. I wonder why increased glaciation (line 385) would lead to more river flow. Build up of glaciers would normally lead to snow accumulation and less net freshwater flux.

5. All in all I believe there is much in the paper and the evidence, and that the paper could be suitable after a revision following these suggestions.

Reviewer #2 (Remarks to the Author):

3D sedimentary architecture showing the inception of an Ice Age by Løseth, H., Dowdeswell, J.A., Batchelor, C.L. and Ottesen, D.

The manuscript presents a study of the change in depositional setting at the transition from the Pliocene "greenhouse climate" to the Quaternary "icehouse" climate. The study is based mainly on high-resolution seismic data and access to I assume cutting samples from wells. There is no doubt that the study reveals morphological features in the North Sea that indicate a dramatic change. The Neogene succession fringing Scandinavia is generally characterized by shoreface-delta progradation each unit rarely exceed more than 200 m thickness. A pattern that have lasted since the initial uplift of Scandinavia in the early Miocene. Also, during this time frame, the sediment flux from Scandinavian were directed southward into the southeastern part of the North Sea. The study by Løseth and colleagues presents new morphological features, i.e., the submarine channel and debris-flow deposits. The size of the features presents in the study is not known from the Neogene succession in the North Sea. Furthermore, the marked westward progradation from Scandinavia as shown by the progradation of the Naust Formation indicate a totally new sediment routing system in the area. The height of the clinoformal package (>500 m) above the Base Naust Formation also indicate a remarkable change in the overall depositional setting of the North Sea Basin.

Therefore, the interpretation presented in the manuscript is likely. The high quality of the seismic data reveals features that must be diagnostic for "green-icehouse" changes and consequently, of interest for a broader audience.

I have attached my comments on the manuscript. There are a few sentences that needs rewriting in order to fulfil a high quality of the manuscript.

Erik Skovbjerg Rasmussen

Reply to review of “3D sedimentary architecture showing the inception of an Ice Age”

Reviewer 1

This is an interesting paper utilizing a unique 3D seismic data set acquired due to the intense oil and gas exploration in the North Sea. It is nice to see this data set being used for paleoclimatic and sedimentological studies of this kind, and this study brings a more classical geological approach to supplement and improve the more general and standard approaches to derive the ice age history based on deep sea sediment records. The ability to draw on the unique seismic data brings different novel aspects of the development and inception of large-scale Northern Hemisphere glaciation to the table. It is therefore a unique and new contribution and despite elements of the story having been available before in various studies, this is a much more comprehensive and potentially important study. From this point of view I am positive to the paper being published, but have some reservations concerning the contents and its ability to communicate to a wide readership which leads me to suggest a relatively major revision before it can be published.

Reviewer 1 was generally very positive about our study and its main findings, but had **four** points of concern that we will address in turn below.

R#1, point 1

1. I am not happy with the use of greenhouse/icehouse jargon in this case. Undoubtedly there was a major expansion of NH ice sheets 2.7-2.6Ma, but to call the late Pliocene period immediately before this a greenhouse world is not very accurate, for several reasons: Since there are lots of evidence for an active NH cryosphere long before this time: e.g. the existence of a Greenland Ice Sheet for several million years, and IRD records that point to NH glaciation in the Arctic and Nordic Seas well back into the late Miocene. One may argue that a final step in the emergence of bipolar major glaciations happened around 2.6Ma, but the world had clearly evolved into an icehouse world long before this event. Existing CO₂-records also indicate relatively low atmospheric concentrations leading into the glacial expansion, hence the greenhouse gas forcing had already been driving climate into a world of active cryospheric influence in both hemispheres.

We understand the reviewer’s comment that the greenhouse/icehouse jargon may be inappropriate for the immediate late Pliocene-Quaternary transition, given the temperature records, changing ocean currents, sea ice expansion and ice on Greenland prior to the Quaternary (e.g. Clotten et al., 2018). Our approach considers a longer time span (late Miocene/Pliocene to Quaternary sediments). In such a perspective, the greenhouse/icehouse terminology is valid because several periods during the late Miocene and Pliocene are described as greenhouse (e.g. Raymo et al. 1996).

However, in light of the reviewer’s comment and to avoid any confusion, we have removed any instances of the term “greenhouse” and changed our terminology to “inception of an ice age” where appropriate. This is valid terminology because we are focussing on the events that characterise the onset of the Quaternary Period, which is commonly referred to as an ice age. We also include a new Supplementary Figure 1 that shows the global d18O record alongside our interpreted changes on the NE Atlantic margin, which should help to place the changes beyond southern Norway at the onset of the Quaternary within their longer-term context.

R#1, point 2

The paper deals with the Northern European and Eastern margin of the North Atlantic. Hence, to refer to this as the North Atlantic is misleading, in particular since the timing of glacial expansion may have been different across the North Atlantic, at least with respect to Greenland. The paper should be more precise in referring to the NE North Atlantic or NW European margin.

This is a helpful comment that will lead to improved precision. We have changed our terminology to refer to the Northeast Atlantic throughout the manuscript.

R#1, point 3

We consider point 3 of reviewer 1 to have three different parts (a) to (c).

(a) The main weakness of the paper, which also impacts on its readability is the lack of discussion of timing and age control. There is a general lacking discussion of the age of the different units and the underlying age control and its uncertainties. To be published, this needs to be in place including giving numeric values of the time of the events and uncertainties (despite a few dates are indicated in Fig. 3).

Whilst well-established numerical ages would certainly improve our paper, few precise ages exist for the Quaternary or pre-Quaternary sediments of the northern North Sea. However, we agree with the reviewer that more information about the ages that *do* exist would make a useful addition to our manuscript.

In light of this comment, we have made four main changes to our paper:

First, we have added a significant amount of text (764 words), and two new references, to the Methods chapter, which summarises our current state-of-the-art understanding of the age of the Quaternary and pre-Quaternary sediments of the northern North Sea. This new text reads:

“Chronological control

There is generally poor chronological control on the Neogene and Quaternary sediments of the northern North Sea. Few long sediment cores have been acquired, and biostratigraphic age assignments from industrial wells are problematic because of large numbers of resedimented fossils and a lack of key indicator micro-organisms³⁸⁻⁴⁰. Here we provide a summary of available age constraints on the Pliocene to Quaternary sediments of the northern North Sea, based on seismic ties and age estimates from published data. These limited age constraints were used to produce a tentative chronology for the Quaternary sediments of the northern North Sea (Supplementary Figure 1). This chronology enables the relative ages of the intra-Naust units to be established regionally and provides a framework into which future age assignments should fit.

Pre-Quaternary ages

The Utsira Formation sand (Fig. 1a) is generally considered to be of Miocene to Pliocene age^{15, 16, 27, 31}. It is difficult to assign precise age estimates to the Utsira Formation because of the abundance of reseeded fossils, which make it difficult to interpret in-situ forms with confidence³⁹. In addition, parts of the Utsira Formation have historically been poorly imaged on seismic data because they have been obscured by massive sand injectites⁴¹. Recent improved age assignments to the Neogene strata of the northern North Sea give slightly younger Pliocene ages for the Utsira Formation^{15, 16}.

The Utsira Formation east is typically assigned an unspecific late Miocene to early Pliocene age in industry completion reports based on biostratigraphic analyses. The stratigraphic position of the Utsira Formation east is above the MMU and below the oldest parts of the Quaternary Naust Formation (Fig. 1). We interpret that the Utsira Formation east sand was deposited after the flooding related to termination of the compression phase⁸ and suggest that these sediments are mainly Pliocene in age.

Quaternary ages

Only one age assignment exists for the Early Quaternary sediments in our study area (60-62°N). The base of the prograding clinoforms of the Naust Formation has been suggested to be younger than 2.6 M yr and older than 1.4 M yr, based on microfossil investigations of industry wells, including wells 34/7-4 and 34/4-7¹⁵ (Fig. 1b). Although this is a large time span, it provides a useful minimum age for these sediments.

Further precise intra-Naust Formation ages do not exist in the northern North Sea, and our estimates (Supplementary Figure 1) are based on seismic ties to the south (Supplementary Figure 2). The top of Unit B (Fig. 1c) has been suggested to be about 1.7-1.6 M yr in age, based on a seismic tie into the study area from the Netherlands¹⁴. The top of Unit B is slightly younger than the top of Sequence 13 of Kuhlmann and Wong⁴², which corresponds with the top of the Olduvai palaeo-magnetic subchron dated to about 1.8 M yr. This age estimate is in broad agreement with the chronostratigraphic framework developed for the central North Sea by Reinardy *et al.*¹⁶ (Supplementary Figure 2), who dated the sediments above the Utsira High using strontium isotopes.

We suggest a tentative age of about 2.2-2.0 M yr for the top of Unit A, because of its stratigraphic position between the base of the Quaternary Naust Formation (~2.6 M yr) and the top of Unit B (1.7-1.6 M yr¹⁴) (Fig. 1c; Supplementary Figures 1 and 2). Clinoforms (i) and (ii) within Unit A were therefore probably deposited during the earliest Quaternary. It is

likely that older sediments existed further landward of clinoform (i) but these were eroded by the Norwegian Channel Ice Stream, which formed the URU (Fig. 1c).

Cliniform (v) within Unit C (Fig. 1c; Supplementary Figure 1) corresponds with horizon R3 of Reinardy *et al.*¹⁶, which has been dated to about 1.5 M yr (Supplementary Figure 2). Subsequent to the deposition of clinoform (v), the deep marine basin connection between Norway and the East Shetland Platform closed as a result of sediment infilling from the surrounding land areas (Fig. 1b), preventing reliable seismic correlations to be made between the northern and central North Sea. However, horizon R4 of Reinardy *et al.*¹⁶, which has been dated to about 1.2 M yr and corresponds to the top of Clinoform Unit 3 of Ottesen *et al.*¹⁴ (Supplementary Figure 2), can be tentatively correlated to approximately clinoform (vi) by counting the prograding clinoform peak reflections either side of the shallow basin between the central and northern North Sea. The age of the top of Naust Unit C, which is the base of the North Sea TMF, is uncertain, but it has been suggested to be as young as 0.5 M yr²⁹.”

Secondly, we have made changes to the text of the main manuscript to provide more information about the ages of the sediments. The changes are shown in red below:

- L55-56: We have added more information about these sediments:

“The sands making up the Utsira Formation on the western flank (Fig. 1a) were produced during the Late Miocene to Pliocene⁹⁻¹⁶. **The flanks were flooded, possibly related to termination of the compression phase⁸, around the end of the Miocene but the uplifted Norwegian mountains remained as nucleation points for the build-up of glaciers.** At about the same time, sandy sediments from Norway formed the Utsira Formation on the basin’s eastern side (Fig. 1a, b).”

- L72: We have added two sentences about the possible ages of these sediments.

“**The top of Unit B has been suggested to have been formed around 1.6-1.7 M yr^{14, 16}. Further precise intra-Naust Formation ages do not exist, but we tentatively suggest the following ages for our horizons based on previous dating work and seismic correlation^{14, 16}: ~2.5 M yr for the oldest preserved Naust (i) clinoform, ~2.0-2.2 M yr for the top of Unit A, and ~1.2 M yr for clinoform (vi) (Methods; Supplementary Figure 1).**”

- L188-193: We change these sentences to include more information about the age of the Utsira Formation:

“First, in the mid-late Miocene **and Pliocene**, sedimentation relating to compression and uplift of Norway and the British Isles was initially in the form of fluvially derived sandy deltaic systems represented by the Utsira Formation **and older** sands (Fig. 3a). **These** Most of these sediments built out into the subsiding northern North Sea Basin **as the Utsira Formation** from the East Shetland Platform in the west **and the Norwegian mainland to the east**. **The upper part of the Utsira Formation, including the Utsira Formation east along the eastern flank (Fig.**

3a), was deposited mainly during the Pliocene after a transgression related to termination of the compression phase⁸. This represents pre-Ice Age greenhouse fluvio-deltaic sedimentation into a mid-latitude marine basin from the surrounding land masses.”

- L349-350: We change the wording of this figure caption to:

“(B) Amplitude map showing the ~~Mid-Late-Miocene-Pliocene~~ to Quaternary infill of the northern North Sea Basin (location in Fig. 1a).”

- L353-358: We add the following information:

“Some of the clinofoms are labelled (i to viii, from oldest to youngest), and the adjusted topbase of Units AB to and C of Ottesen *et al.*^{13,14} and Batchelor *et al.*²⁰ are shown. Regional ties and estimates suggest the following ages: top Unit A ~2.2-2.0 M yr, top Unit B ~1.7-1.6 M yr, clinofom (v) ~1.5 M yr, and clinofom (vi) ~1.2 M yr (Methods; Supplementary Figure 1). Pink fill is an early Quaternary deltaic unit that prograded to the north-east from the East Shetland Platform¹⁴. Yellow fill is the Utsira Formation east (of probable Pliocene age), which was deposited prior to the glacial debris-flows. Red circles are wells. (C) Seismic profile showing the Pliocene ~~Late-Miocene~~ to Quaternary infill of the northern North Sea Basin (location in Fig. 1b).”

Thirdly, we have added a Supplementary Figure that places our stratigraphy within a chronological framework for events in the North Sea. We have indicated (in the main text, figure caption, and use of dashed lines and question marks in the Supplementary Figure) that these correlations are tentative, given the lack of well-established dates for these sediments.

Fourthly, we have added a second Supplementary Figure that shows how our horizons correlate with those of previously published work, including their tentative dates, in the central North Sea.

R#1, point 3(b)

It would also help for the general readability if these events were placed into the global and regional deep sea evidence from O-isotopes and IRD-records.

We have added a new Supplementary Figure 1 that places our study into a global and regional context by showing our interpreted horizons and units alongside the global d18O record.

R#1, point 3(c)

A discussion on the possible mechanisms behind the continued expansion of ice sheets/ice streams onto the shelves, e.g. general climatic and oceanographic causes vs changes in ice sheet dynamics due to the changing topography of the mountains and coastal regions due to glacial erosion. Here a mentioning of eustatic and isostatic influences would also be appropriate. I think the paper misses an opportunity to have a wider impact due to these deficiencies.

In light of this comment, we have added the following sentences to the main text to discuss the possible mechanisms for the expansion of the ice sheet onto the continental shelf at the start of the Quaternary. These sentences read:

“Ice-sheet expansion along the Northeast Atlantic margin at the beginning of the Quaternary was driven by large-scale climatic and oceanographic forcing, which was the culmination of a general cooling trend through the Late Cenozoic³³ (Supplementary Figure 1). Southern Norway was one of the key nucleation centers for the Northern Hemisphere ice sheets during the earliest Quaternary because of its mountainous terrain and proximity to a moisture

source²⁴. The existence of a deep basin in the northern North Sea during this time^{13, 14} provided accommodation space for the delivery of large volumes of glacial sediment across the relatively narrow continental shelf to the palaeo-shelf break.”

“Continued isostatic subsidence of the North Sea Basin through the Quaternary, combined with erosion and isostatic uplift of the surrounding land areas⁴⁻⁸, facilitated high rates of glacier-derived sediment deposition delivered through expanding glacial valleys. After about 1 M yr, the infilling of the North Sea Basin, combined with Northern Hemisphere climatic deterioration and ice-sheet expansion linked to the Mid-Pleistocene Transition³⁶, enabled confluence of the Scandinavian and British-Irish Ice Sheets, initiation of the Norwegian Channel Ice Stream, and the shifting of the focus of glacial sediment deposition into the Norwegian Sea²⁰.”

R#1, point 4

The caption of fig. 3 includes discussion and arguments which should have appeared in the main text.

We agree with Reviewer 1 and have removed this information from the figure caption. We note that this information is included on L124-125 of the main text.

I wonder why increased glaciation (line 385) would lead to more river flow. Build up of glaciers would normally lead to snow accumulation and less net freshwater flux.

Reviewer 1 correctly claims that snow accumulation generally leads to reduced freshwater flux to the sea. However, in areas of large temperature variation between winter and summer snow will accumulate mainly during the winter. This snow is a “water reservoir” that will be tapped into during summer when higher temperatures allow melting. Therefore, the freshwater flux is periodically higher when melting snow adds to rainwater discharge. We have changed the following in the main text to make this clearer:

L122-125: We interpret this major Sunnfjord channel, whose preserved upper arms point towards the Sunnfjord and Nordfjord systems on mainland Norway (Fig. 1a), as a submarine product of enhanced **summer seasonal** meltwater and sediment delivery after the initial build-up of glaciers and ice caps in the Norwegian mountains.

L384-386: **(B)** Around the onset of the Quaternary Period, glacial debris-flows record the first advance of grounded ice to the palaeo-shelf break beyond Sognefjord. Sunnfjord channel was probably formed during this time. ~~This formed a trough-mouth fan, composed of numerous stacked glacial debris flows, along the eastern side of the northern North Sea Basin. This formed a through mouth fan, composed of numerous stacked glacial debris flows, along the eastern side of the North Sea Basin. The formation of Sunnfjord channel beyond Sunnfjord and Nordfjord around this time may have been linked to an increase in river discharge as a consequence of the expanded glaciers in southern Norway, and an early Quaternary fluvial delta built up beyond the East Shetland Platform¹⁴.~~

All in all I believe there is much in the paper and the evidence, and that the paper could be suitable after a revision following these suggestions.

We thank Reviewer 1 for their thorough and helpful review of our manuscript, which has greatly improved the paper.

Reviewer 2

The manuscript presents a study of the change in depositional setting at the transition from the Pliocene “greenhouse climate” to the Quaternary “icehouse” climate. The study is based mainly on high-resolution seismic data and access to I assume cutting samples from wells. There is no doubt that the study reveals morphological features in the North Sea that indicate a dramatic change. The Neogene succession fringing Scandinavia is generally characterized by shoreface-delta progradation each unit rarely exceed more than 200 m thickness. A pattern that have lasted since the initial uplift of Scandinavia in the early Miocene. Also, during this time frame, the sediment flux from Scandinavian were directed southward into the southeastern part of the North Sea. The study by Løseth and colleagues presents new morphological features, i.e., the submarine channel and debris-flow deposits. The size of the features presents in the study is not known from the Neogene succession in the North Sea. Furthermore, the marked westward progradation from Scandinavia as shown by the progradation of the Naust Formation indicate a totally new sediment routing system in the area. The height of the clinoformal package (>500 m) above the Base Naust Formation also indicate a remarkable change in the overall depositional setting of the North Sea Basin. Therefore, the interpretation presented in the manuscript is likely. The high quality of the seismic data reveals features that must be diagnostic for “green-icehouse” changes and consequently, of interest for a broader audience.

I have attached my comments on the manuscript. There are a few sentences that needs rewriting in order to fulfil a high quality of the manuscript.

Erik Skovbjerg Rasmussen

Reviewer 2 was very positive about our manuscript, but had a few comments and suggestions to improve the text. Descriptions of our changes are listed below:

- L24. We have changed “detail” to “details”

- L58 to 62: We have rewritten these sentences as suggested by the reviewer. They now read: “Here, the Utsira Formation east is **bounded at its base** ~~underlain~~ by the Mid-Miocene Unconformity (MMU; Fig. 1c), an unconformity on the entire eastern basin margin that dips smoothly westwards showing reflection truncation, onlap and downlap^{17, 18}. The Utsira Formation is overlain by **the a-down-lapping surface that separates the sand from the overlying**-glacially influenced prograding shelf clinoform sediments of the Quaternary Naust Formation (Fig. 1c).”

- L89. Seismic records indicate that the sands **unit hasve** a well-defined top...

- L92. **Channels are widening towards the east – this is unusual for a delta – explain!**

This is the marine part of a delta that was downlapping and thinning to the west. The seafloor was probably flattening out westward and the gravitational energy was decreasing. Also, several channels approach a point source in the east that also may indicate reduced energy. If the deposited sediments were very rich in sand and lean in mud this may have prevented the sediments from going into a turbiditic mode. This can explain why the channels become narrower towards the deeper part of the basin. Because this is a description section, we do not add further discussion of this point to the text here.

However, we highlight the obvious narrowing of the channels to the west by changing the text to read: “Individual channels are usually less than 800 m wide, **become narrower to the west, and** can be traced for up to 30 km.”

- L95. Ref #2 would like to have a reference here.

The assertion that a valley system existed in this location prior to erosion of the fjord by ice is based on our interpretation of the Utsira east as a fluvially derived delta. To our knowledge, there are no publications that point to the existence of a valley here during the Miocene or Pliocene. To avoid confusion, we have removed the last part of this sentence:

“..mountainous interior of Norway (Fig. 1a), ~~which was a fluvial valley prior to glaciation.~~”

- L358. Ref#2 says that he can't see where Fig. 1c is indicated within Fig. 1b.

We have checked this. The location of the seismic section in Fig 1c is shown clearly as a dotted and labelled red line within Fig. 1b, so we have not made any changes in response to this comment.

REFERENCES:

- Clotten, C., Stein, R., Fahl, K., Schreck, M., Risebrobakken, B., & De Schepper, S. (2019). On the causes of Arctic sea ice in the warm Early Pliocene. *Scientific reports*, 9(1), 1-8.
- Evans, D. (Ed.). (2003). *The Millennium Atlas: Petroleum Geology of the Central and Northern North Sea; [a Project of the Geological Society of London, the Geological Survey of Denmark and Greenland and the Norwegian Petroleum Society]*.
- Løseth, H., B. Raulline, and A. Nygård. "Late Cenozoic geological evolution of the northern North Sea: development of a Miocene unconformity reshaped by large-scale Pleistocene sand intrusion." *Journal of the Geological Society* 170.1 (2013): 133-145.
- Kuhlmann, Gesa, and Theo E. Wong. "Pliocene paleoenvironment evolution as interpreted from 3D-seismic data in the southern North Sea, Dutch offshore sector." *Marine and Petroleum Geology* 25.2 (2008): 173-189.
- Raymo, M. E., Grant, B., Horowitz, M., & Rau, G. H. (1996). Mid-Pliocene warmth: stronger greenhouse and stronger conveyor. *Marine Micropaleontology*, 27(1-4), 313-326.
- Sømme, T. O., Jackson, C. A. L., & Vaksdal, M. (2013). Source-to-sink analysis of ancient sedimentary systems using a subsurface case study from the Møre-Trøndelag area of southern Norway: Part 1—depositional setting and fan evolution. *Basin Research*, 25(5), 489-511.

REVIEWERS' COMMENTS:

Reviewer #1 (Remarks to the Author):

I thank the authors for careful attention to my remarks. The sections on chronology and the terminology changes are fine, given the constraints of available data. The paper is now significantly improved, and I am happy to recommend it for publication.